# Hybrid Microcapsules for Encapsulation and Controlled Release of Rosemary Essential Oil

**DOI:** 10.3390/polym15040823

**Published:** 2023-02-07

**Authors:** Doha Berraaouan, Kamal Essifi, Mohamed Addi, Christophe Hano, Marie-Laure Fauconnier, Abdesselam Tahani

**Affiliations:** 1Physical Chemistry of Natural Substances and Process Research Team, Laboratory of Applied Chemistry and Environment (LCAE-CPSUNAP), Faculty of Sciences, Université Mohamed Premier, BV Mohammed VI BP 717, Oujda 60000, Morocco; 2Laboratoire d’Amélioration des Productions Agricoles, Biotechnologie et Environnement (LAPABE), Faculty of Sciences, Université Mohamed Premier, BV Mohammed VI BP 717, Oujda 60000, Morocco; 3Laboratoire de Biologie des Ligneux et des Grandes Cultures, INRAE USC1328, Campus Eure et Loir, Orleans University, 28000 Chartres, France; 4Laboratory of Chemistry of Natural Molecules, Gembloux Agro-Bio Tech, University of Liège, 5030 Gembloux, Belgium

**Keywords:** encapsulation, hybrids, clay, sodium alginate, rosemary essential oil, controlled release

## Abstract

The foremost objective of this work is to assess the microcapsules composition (polymer-based and polymer/clay-based) effect, on the release of rosemary essential oil into w/o medium and evaluate their antioxidant activity. Calcium alginate (CA) and calcium alginate/montmorillonite hybrid (CA-MTN) microcapsules were developed following an ionotropic crosslinking gelation and were used as host materials for the encapsulation of rosemary essential oil. The unloaded/loaded CA and hybrid CA-MTN microcapsules were characterized by Fourier transform infra-red (FT-ATR) spectroscopy, thermal analysis (TGA), scanning electron microscopy (SEM) and DPPH assay. The evaluation of the microcapsule’s physicochemical properties has shown that the clay filling with montmorillonite improved the microcapsule’s properties. The encapsulation efficiency improved significantly in hybrid CA-MTN microcapsules and exhibited higher values ranging from 81 for CA to 83% for hybrid CA-MTN and a loading capacity of 71 for CA and 73% for hybrid CA-MTN, owing to the large adsorption capacity of the sodic clay. Moreover, the hybrid CA-MTN microcapsules showed a time-extended release of rosemary essential oil compared to CA microcapsules. Finally, the DPPH assay displayed a higher reduction of free radicals in hybrid CA-MNT-REO (12.8%) than CA-REO (10%) loaded microcapsules. These results proved that the clay–alginate combination provides microcapsules with enhanced properties compared to the polymer-based microcapsules.

## 1. Introduction

Essential oils are liquids with concentrated molecules resulting from the metabolism of a plant. They are very popular in the cosmetics, perfumery, food preservation, and pharmaceutical sectors [1]. The oxidative stress that harms biological molecules can be diminished by essential oils, and many degenerative diseases, including, diabetes, cardiovascular diseases, and neurological disorders, can also be successfully treated [2]. Despite the great biological potential that essential oils currently possess, they are under-utilized when their applications become difficult due to certain physicochemical characteristics provided by the compounds present in their compositions [3]. It is difficult to use essential oils in aqueous environments because they are generally highly volatile at room temperature, easily oxidized when exposed to oxygen and light, and above all have a high lipophilicity [4]. Aqueous solubilization and preservation of bioactive substances present in essential oils are crucial for their application, which requires the use of a system to promote their dispersion in aqueous media and regulate their release. These issues can be resolved through the incorporation of essential oils into various systems. Included in these systems are emulsions, beads, bioactive films, capsules, liposomes, nanocarriers, and inclusion complexes [5]. Encapsulation systems have attracted great interest in recent decades with a view to achieving an efficient drug delivery process [6]. Particular attention has been paid to finding an inert carrier where the drug is dispersed or incorporated. Recently, the preparation, characterization and application of controlled drug delivery systems prepared from biopolymer compounds has become popular owing to their distinctive properties such as a high encapsulation efficiency, biodegradability, nontoxicity, and controlled release characteristics [7]. The encapsulation of essential oils or their isolated components has been investigated by some researchers where they worked on the possibilities of using essential oils as a drug vehicle in delivery systems for lipophile drugs. Similarly, Rodriguez et al. [8] reviewed the possible strategies for essential oils encapsulation and showed the benefits of polymeric nanoencapsulation [9].

Out of the many biopolymers used for encapsulation purposes, alginate is widely used due to its biosafety, mild gelation properties, environmentally friendly [10], hydrophilicity, as well as its ability to be modified or combined with other biomaterials [11]. It is a hydrophilic polysaccharide, composed of two monomeric structures: β-D mannuronic acid (M) and α-L-guluronic acid (G) which are arranged in MM or GG blocks interspersed with MG blocks [12]. This structure has the unique property of forming water-insoluble calcium–alginate gel through ionotropic gelation with divalent cross-linking salt such as calcium chloride [13]. But the low efficiency of encapsulating water-soluble substances in alginate is one of the problems in developing a controlled delivery system [14]. In addition to the many advantages of polymeric encapsulation, it suffers from low strength. This disadvantage can be overcome by using natural clays as a filler; these systems are called hybrid capsules. Clay minerals have been proposed as fundamental constituents of various modified carriers and have different purposes. Among the promising clay materials, montmorillonite (MTN), which belongs to the smectites family, is of particular interest. The smectite clays are a layered aluminosilicate derived from the assembly of tetrahedral [SiO_4_]^4−^ and octahedral [AlO_3_(OH)_3_]^6−^ sheets. Hybrid systems on the basis of an organic and inorganic materials amalgam have been developed to solve the problem of low chemical stability, low mechanical strength and the erratic release of the encapsulated substances. They are very attractive for diverse applications due to the features afforded by the combination of the organic and the inorganic materials. Among its popular applications is active packaging. In this technology, materials loaded with active agents as essential oils are incorporated to the packaging to inhibit food degradation by oxidation, contamination and many other factors [15,16]. Research work has dealt with nanohybrids valued as possible containers for the encapsulation of rosemary essential oil using pectins and halloysite nanotubes HNTs (a rare type of kaolinite) [17]. It was observed that the release of rosemary essential oil was much slower in the nanohybrid material than the release of the same essential oil blended into the pectin. In addition, the release from the composite filled with the clay shows multiple steps, a rapid diffusion in the first stage, followed by a decreased rate of release in the remaining stages. The antimicrobial activity showed a mold formation in the pectin composites after two weeks of storage, but was not detected in the hybrid composites even after three months. This shows that to elaborate efficient active packaging, it is necessary to entrap the active agent for as long as possible instead of enabling its fast release and that this characteristic can be provided by clay filling.

The preparation of a polymeric layered silicate composite offers the possibility of enhancing the properties of individual components. Different biocompatible and biodegradable polymers are suitable drug carriers that can release components at a constant rate [18]. To improve the active substance entrapment efficiency and thereby modulate the release of the active substance, it is desirable to incorporate materials such as clays. Thus, the alginate–clay composite formed would decrease the active substance release by increasing its loading capacity in the composite matrix. However, one of the main challenges facing the scientific community is the development of appropriate matrices for essential oils encapsulation and protection, as well as their controlled release under specific stimuli, which we call “smart carriers” [19,20].

The main objective of the present study is the elaboration of composites prepared in situ by ionotropic gelation of sodium alginate and the combination of the latter with natural sodium bentonite from the eastern region of Morocco for the encapsulation of rosemary essential oil (REO) and the pathway release as a function of microparticles formulation and REO concentration. The microcapsules were characterized by Fourier transform infrared spectroscopy (ATR-FTIR), thermogravimetric analysis (TGA), and scanning electron microscopy (SEM). In addition, the release profile of the composites in an aqueous medium was studied in vitro at an ambient temperature of 20 °C and the ability of the microcapsules to scavenge the DPPH free radicals was also examined.

## 2. Materials and Methods

### 2.1. Materials

Sodium alginate (SA) (15% loss during drying at 105 °C, 30% ignition residue and 0.004% heavy metal content) was purchased from Panreac Quimica (Barcelona, Spain). The polymer presented intrinsic viscosities of 1.03 × 10^3^ mL/g and 5.39 × 10^3^ mL/g in 0.1 M sodium chloride and distillate water, respectively, measured using an Ubbelohde capillary viscometer at 25 °C. The values of the typical molecular weight are 5.48 × 10^4^ g/mol and 3.38 × 10^5^ g/mol for 0.1 M sodium chloride and distillate water, respectively. In addition, the M/G ratio was determined using infrared spectra and was equal to 1.37. The raw bentonite was taken from the Azzouzet deposit (Nador, Morocco), previously characterized [21], purified and exchanged before the experiment to obtain a sodic clay. The main parameters of the raw and purified/exchanged clay are shown in Table 1.

*Rosmarinus officinalis* L. essential oil (REO) was obtained from a distillation unit in Jerrada (eastern region of Morocco) and was stored at −1 °C until examination. Tween 80 (density of 1.06 and viscosity of 300–500 mPa⋅s at 25 °C) was provided by Panreac Quimica (Barcelona, Spain) and calcium chloride (powder, 97% with a molar mass of 110.99 g/mole) was acquired from Riedel-de-Haën (Seelze, Germany).

### 2.2. Preparation of Rosemary Essential Oil Loaded Microcapsules

The preparation of the loaded microcapsules followed two steps as in Figure 1. Firstly, the preparation of emulsions containing alginate/montmorillonite/REO and alginate/REO was carried out, followed by the preparation of the microcapsules through the addition of those emulsions into a gelling bath.

For calcium alginate CA microcapsules, a sodium alginate solution (1% *w*/*v*) was dissolved in distillate water while being stirred magnetically (350 rpm) at room temperature (20 °C). An oil/water emulsion was formed by mixing REO with sodium alginate solution and left to stir overnight to reach a final concentration of 1, 2 and 3% of rosemary essential oil. For the CA-MTN hybrid microcapsules, a dispersion of sodium bentonite MTN (4%) in distillate water was left to stir overnight. Different concentrations of rosemary essential oil (1, 2 and 3%) were added to the sodium bentonite dispersion and left to stir overnight for the maximum adsorption of essential oil on the bentonite. The sodium MTN/REO emulsion was then blended with sodium alginate solution with a 1:2 ratio and left to stir for 5 h. The amount of alginate solution and bentonite dispersion used in hybrid CA-MTN microcapsules were calculated to obtain the same final essential oil concentration as in CA microcapsules. The CA and CA-MTN hybrid microcapsules were generated by adding the emulsions dropwise to calcium chloride solution (0.1 M) for 60 min under magnetic stirring. The microcapsules were retrieved through filtration, repeatedly rinsed with distilled water and then put into storage at 4 °C.

### 2.3. Identification of Rosemary Essential Oil Chemotype

The gas chromatographic-MS analysis of *Rosmarinus officinalis* L. essential oil was performed via a Hewlett-Packard 6890 gas chromatograph interfaced with a Hewlett-Packard mass selective detector (Agilent Technologies, Santa Clara, CA, USA). An HP1 fused silica column (Phynel-methyl Siloxane 30 m × 0.25 mm i.d., film thickness 0.25 μm) was used. An interface temperature of 280 °C was used for the gas chromatography parameters, while for the mass spectrometry parameters an interface temperature of 250 °C. The MS source temperature was of 200 °C, the ionization energy and the ionization current were of 70 eV and 2A, respectively [22]. The carrier gas was helium and the flow rate along the column was 1.4 mL/min.

### 2.4. Characterization of Microcapsules

#### 2.4.1. Attenuated Total Reflection Fourier Transform Infrared (ATR-FTIR)

The attenuated total reflectance-Fourier transform infrared (ATR-FTIR) analysis was carried out using a Shimadzu Jasco 4700-ATR spectrophotometer (Kyoto, Japan) to determine the functional groups in the rosemary essential oil, as well as unloaded and loaded CA and CA-MTN hybrid microcapsules in the wavelength region between 400 and 4000 cm^−1^. The obtained spectra for each sample were affected by averaging 32 scans at a resolution of 4 cm^−1^.

#### 2.4.2. Thermogravimetric Analysis (TGA)

The thermogravimetric analysis was carried out via a SHIMADZU TA-60WS thermal analyzer (Kyoto, Japan) with an initial sample mass of 22 and 31.8 mg for CA and CA-MTN hybrid microcapsules, respectively, in an alumina sample holder at a heating rate of 20 °C/min in an N_2_ atmosphere.

#### 2.4.3. Particle Size and Morphology

The microcapsules sizes and shapes are related to their circularity, which indicates how closely the shape of the microparticle resembles a circle. Its value ranges from 0 to 1; the latter indicates the perfect sphere. The size was estimated using an Olympus SZ Stereomicroscope (Japan), equipped with a digital camera (D5000 Color Video Camera, Nikon (Tokyo, Japan), and the shape was characterized by aspect ratio and shape factor using an Image J Software program (NIH). The morphology analysis of the microcapsules was conducted by scanning electronic microscopy (SEM), realized in JNCASR (Bangalore, India), using a ZEISS Gemini model. Samples were dried using a critical point dryer and then fixed into a carbon tape for SEM imaging.

#### 2.4.4. Loading Capacity and Encapsulation Efficiency

The amount of encapsulated rosemary essential oil was determined using a Rayleigh UV–VISIBLE 1800 spectrophotometer (Model Jasco 560, Pekin, China), following the method described by El Hosseini [23] with some modifications; 200 mg of the microcapsules were poured into a beaker containing distilled water and Tween 80 (1% *w*/*v*) then left to stir. The absorbance of the obtained solution was measured at 257 nm. The loading capacity and encapsulation efficiency were determined at the final concentration measured at the end of the release study and calculated using the rosemary essential oil calibration curve in the aqueous solution (1% Tween 80). The values were calculated following Equations (1) and (2) [24,25] for loading capacity and encapsulation efficiency, respectively.
(1)$$\;LC\;\left(\%\right)=\frac{\mathrm{Weight}\;\mathrm{of}\;\mathrm{essential}\;\mathrm{oil}\;\mathrm{in}\;\mathrm{microcapsules}}{Weight\;of\;microcapsules}\times 100\;$$
(2)$$\;\;\;EE\;\left(\%\right)=\frac{\mathrm{Weight}\;\mathrm{of}\;\mathrm{essential}\;\mathrm{oil}\;\mathrm{in}\;\mathrm{microcapsules}}{\mathrm{Weight}\;\mathrm{of}\;\mathrm{essential}\;\mathrm{oil}\;\mathrm{added}}\times 100$$

### 2.5. Determination of Antioxidant Activity

The antioxidant activity of rosemary essential oil loaded and unloaded microcapsules was assessed using 1.1-diphenyl-2-picrylhydrazyl (DPPH), a free radical, following the procedure outlined by Ling et al. [26]. Briefly, a 0.1 mM solution of 1.1-diphenyl-2-picrylhydrazyl (DPPH) radical solution in 90% ethanol was prepared and 2.5 mL of this solution was mixed vigorously with different REO concentrations (25–7500 μg/mL in ethanol) and a mass of 100 mg of the loaded and unloaded microcapsules. After 30 min of incubation in the dark at room temperature, absorbance (A) was measured at 517 nm using a Rayleigh UV–VISIBLE 1800 spectrophotometer (Model Jasco 560, Pekin, China). The percentage of the radical scavenging capture (RSC) was calculated based on the following Equation (3), where A_cont_ and A_sample_ are the absorbance values of the control and the sample, respectively.
(3)$$RSC\left(\%\right)=\frac{Acont-Asample}{Acont}\times 100$$

### 2.6. REO Release Kinetics

The REO release study was carried out using a method described by El Hosseini et al. [23]. The REO-loaded microcapsules (250 mg) were placed in a flask containing 30 mL of the release medium (1% Tween 80 in distilled water). The suspensions were slowly stirred at ambient temperature (20 °C). At predetermined time intervals, samples from the release medium were taken. The REO concentration was determined by spectrophotometry at 257 nm using a Rayleigh UV–VISIBLE 1800 spectrophotometer (Model Jasco 560, Pekin, China). The concentration of rosemary essential oil in the release medium at different sampling times was determined using a calibration curve of free REO in 1% Tween 80. The cumulative percentage of the REO release (%CR) was calculated using Equation (4) [27]. To forecast and correlate the REO release behaviour from both microparticles, it is imperative to include the proper model. Therefore, the data from the release study were adjusted to an established empirical model by Korsmeyer and Peppas using Equation (5) [23].
(4)$$\%CR=\overset{t}{\underset{t=0}{\sum }}\frac{M_{t}}{M_{0}}\times 100$$
(5)mtm∞=k×tn
where **^m_t_^/_m_∞__** is the percentage of the drug released at time t and for all time; k is the description of the macromolecular network system and n is the release exponent indicating the release mechanism. A linear form of the equation can be acquired through plotting ln (**^m_t_^/_m_∞__**) against ln(t), whose linear coefficient is k and whose angular coefficient is n. The determined n value is used to discover which of the three mechanisms can describe the release behaviour: the n value is less than 0.43; the release follows the Fickian law and the n value is between 0.43 and 0.85; the non-Fickian release mechanism is established, and when the n value is greater than 0.85 it indicates the case II of transport release with a polymer chain relaxation [28]. Using OriginPro 2018 software, we calculated the squared correlation coefficient (R^2^ was used to confirm the accuracy of the model).

## 3. Results

### 3.1. Identification of Rosemary Essential Oil Chemotype

Essential oil of *Rosmarinus officinalis* L. from the eastern region of Morocco was characterized by gas chromatography-mass spectroscopy (GC-MS) [22]. As can be seen in Table 2, 32 compounds were totally identified, representing 99.9% of the total oil content. The most abundant compounds of essential oil are mainly concentrated in both groups, monoterpenes hydrocarbons, and oxygenated monoterpenes. The main compounds are 1.8-cineole (29.71%), α-thujene (14.17%), camphor (13.09%), β-pinene (9.94%), and camphene (6.46%). The high ratio of these compounds was previously revealed by some researchers [29,30] who worked on the same oil and according to the literature, the essential oil of *Rosmarinus officinalis* L. used in this work can be classified as a 1.8-Cineol chemotype. The volatility of these substances restricts their use even though they can slow down microbial growth or free radicals. Thus, encapsulating them in microcapsules is undoubtedly a better way to delay their volatilization, which can increase the range of their application.

### 3.2. Characterization of Microcapsules

#### 3.2.1. Attenuated Total Reflection Fourier Transform Infrared (ATR-FTIR)

The attenuated total reflection Fourier transform infrared was performed on REO, REO-loaded microcapsules and non-loaded microcapsules to identify the characteristic bands of the chemical structures. The obtained spectra are shown in Figure 2. The REO spectrum presents characteristic bands from 1030 to 1135 cm^−1^ assigned to out-of-plane C-H wagging vibrations from terpenoids. The stretching vibrations of C–O present in the carbohydrates correspond to peaks from 1190 to 1035 cm^−1^. The spectral region from 1700 to 1500 cm^−1^ is associated with C=C bending. The peaks attributed to the C–H bending (aliphatic domain) appear in the 2900 cm^−1^ frequency. Carboxylic acid (O–H) stretching has characteristic absorption between 3000–2500 cm^−1^. The FTIR spectrum of the pure rosemary essential oil shows the expected characteristic C–H stretch (~2900 cm^−1^), C=O stretch (~1700 cm^−1^), broad O–H stretch (~3400 cm^−1^), and C–O stretch (~1100 cm^−1^) of terpenoid components.

The “fingerprint” characteristic bands of sodium alginate include the spectral region from 800 to 1000 cm^−1^ and corresponds to the C–C stretching of the alginate skeleton and deformation mode [31]. The bands that occur at 1417 and 1617 cm^−1^ correspond to its carboxylic groups, symmetric and asymmetric stretching, respectively [32], and have been observed in all the capsule spectra. The broad band around 3600 cm^−1^ in the ATR-FTIR spectrum of hybrid CA-MTN microcapsules is associated with the Al–OH and Mg–OH vibrations. The intensive peak at a wave number of 1036 cm^−1^ represents the Si–O stretching vibrations, which are mainly associated with montmorillonite clay. The peak at 1640 cm^−1^ is attributed to the bending vibration of water that is physically adsorbed [33]. The broad band at 3400 cm^−1^ corresponds to the O–H stretching of interlayer adsorbed water. Regarding the spectra of the REO-loaded microcapsules, some peaks characteristic of rosemary essential oil aromatic domain (2600–3200 cm^−1^) appeared on the spectra of loaded microcapsules and do not appear on the non-loaded microcapsules. Some peaks typical of calcium alginate (1417–1617 cm^−1^) were intensified compared to the non-loaded microcapsules suggesting interactions between the polymer, clay and rosemary essential oil. In the ATR-FTIR spectra, the peak of sodium alginate carboxyl anions was shifted from 1627 cm^−1^ for CA-MTN microcapsules to 1636 cm^−1^ for CA-MTN-REO and from 1630 cm^−1^ to 1651 cm^−1^ for CA and CA-REO, respectively. This can be explained by the REO incorporation. Our presumption is that REO molecules were adsorbed on the clay surface and its interlayer space which induced interaction and the creation of new bonding. In the work of Volic et al. [31], the combination of soy protein, sodium alginate and thyme oil created an aggregation of the soy protein involving a hydrogen bonds disruption.

#### 3.2.2. Thermogravimetric Analysis (TGA)

Thermal analysis of the unloaded and loaded microcapsules showed the weight loss of microcapsules related to moisture. According to Wang et al. [34], alginate polymer has two sorts of water; unbounded water whose thermal event occurs below 100 °C and bounded water whose thermal event occurs above 100 °C. Thermogravimetric studies have shown that the stability of several antioxidants such as essential oils decreases with the increase in temperature [35]. The thermograms (Figure 3) present the weight loss pattern of CA, CA-MTN microcapsules, CA-REO and CA-MTN-REO microcapsules. REO is sensitive to heat and therefore the thermal property was carried out up to 150 °C.

All of the microcapsules show thermal stability up to 42 °C. A weight loss then takes place and slows at a constant rate to 83 °C. This weight decrease of 30% is merely due to the REO present on the capsule’s surface and a part of the external moisture losses [36]. Between 83 and 117 °C, a fast weight loss appears in both loaded microcapsules compared to the non-loaded microcapsules. Until 117 °C, the evaporation of the outer mist and the diffusion of the REO within the core capsule occur faster in CA microcapsules than in CA-MTN hybrid microcapsules. At that stage, the loss is important and exceeds 48%. The study continues until total decomposition. The results have shown that the presence of montmorillonite clay improved the thermal stability of the hybrid microcapsules compared with CA microcapsules. This improvement in the thermal stability is related to the reduced motions of alginate networks following the clay filling which slows the diffusion of the inner particles through the capsule membrane [37]. Therefore, REO could be better preserved in the CA-MTN hybrid microcapsules for encapsulation purposes.

#### 3.2.3. Particle Size and Morphology

The size and morphological characterization of the microcapsules are important because they can have an impact on the physicochemical properties such as water sorption and release of the encapsulated agent, as well as the aesthetic quality which could be a desirable feature for pharmaceutical and food products [38]. Because the microcapsules were generated by ionotropic gelation of the emulsion through a needle of 1 mm diameter into a calcium gelling bath under magnetic stirring they differ in size and shape. The average size of hydrated microcapsules is shown in Figure 4 and Table 3 and were of 1.71 ± 0.1 mm for CA and 1.64 ± 0.5 mm for CA-MTN. The size of the microcapsules indicates that they are micrometric. The shape and morphology of the microcapsules were determined from the aspect ratio and circularity factor. The circularity values equal or closely equal to zero indicate a perfect sphere. The values are of 0.92 and 0.94 for CA and CA-MTN microcapsules, respectively, and show the morphological characteristics of a sphere.

The SEM micrographs (Figure 5) show that the CA microcapsule (Figure 5a) has an irregular form and a wrinkled surface. High magnificence SEM micrographs; 1220× and 1370× (Figure 5c,e), we can observe that an alignment of lines attributed to the polymer chains and its external surface is covered with a network of fissures and cracks. However, the hybrid CA-MTN (Figure 5b) microcapsules show an improved spherical form, a more compact surface and less irregularity. On the micrographs Figure 5d,f, the polymeric chains in the hybrid CA-MTN microcapsules are associated with the montmorillonite aggregates to form a crumpled network. The hybridation improved the aesthetic aspect of the microcapsules and the polymer matrix strength by filling its interstitial voids and therefore increasing the “time hosting” of the encapsulated agent in the hybrid CA-MTN microcapsules.

#### 3.2.4. Loading Capacity and Encapsulation Efficiency

The results in Table 4 showed that the loading capacity and encapsulation efficiency are influenced by the essential oil content. The loading capacity increased when we increased the rosemary essential oil concentration (3%), 71% and 73% for CA and hybrid CA-MTN microcapsules, respectively, whereas the encapsulation efficiency decreases with an increase in the essential oil amount. The loading capacity depends on the weight of the REO in the microcapsules and the weight of the microcapsules. Our assumption is that the clay filling densifies the microcapsule’s matrix network, creating more space for the oil to fit and consequently improving the loading capacity by 2% for the 3% REO concentration. However, the encapsulation efficiency decreases with the increase in essential oil concentration for CA microcapsules. This can be explained by the fact that the polymer network is insufficient to host a higher amount of REO, suggesting a saturation of the material capacity. While for the hybrid CA-MTN microcapsules, thanks to the combination alginate-montmorillonite, they showed a better entrapment of 2% REO concentration equal to 94% compared to CA microcapsules, which is equal to 81%. These findings are supported by El Hosseini et al. [23] who worked on the encapsulation of sunflower in sodium alginate microparticles. The results indicated the highest encapsulation efficiency for the lowest sunflower concentration and the highest loading capacity for the highest sunflower concentration. Piornos et al. [39] reported that the encapsulation efficiency is related to the system of encapsulation strength, which can explain the improvement from adding the montmorillonite clay in this work.

### 3.3. Determination of the Antioxydant Activity

The microcapsules’ ability as antioxidants was tested. In the present study, the elaborated unloaded microcapsules, CA and CA-MNT, as well as the loaded microcapsule CA-MNT–3% REO were tested for their capability to reduce the free radical 1.1-diphenyl-2-picrylhydrazyl (DPPH); the REO was also evaluated for its antioxidant activity. The results of the scavenging activity performed are highlighted in Figure 6. As can be seen from these results, the unloaded microcapsules showed the lowest scavenging activity, presenting a value of 8.6% and 4.8% for CA and hybrid CA-MNT, respectively. These results can be explained by the fact that calcium alginate has the ability to react with the free radical compared to bentonite clay, which increases the free radical inhibition. Furthermore, there was a slight increase in the free radical scavenging activity of CA-REO compared to the control CA (from 8.6 to 10%); on the other hand, the CA-MNT-REO exhibited the highest scavenging activity, reaching 12.8%, which was ten times higher than that of CA-MNT (4.8%). 

Based on the literature data, the same results were found by Singh et al. [40] reporting that the chitosan-gelatin-REO microcapsules exhibited a higher antioxidant property than the unloaded microcapsules. Similarly, Teixeira-Costa et al. [41] revealed that the essential oil encapsulated in the system chitosan/alginate polyelectrolyte complexes inhibits greatly compared to the control polyelectrolyte complexes. These results are mainly due to the presence of oxygenate monoterpenes and hydrocarbon monoterpenes which have significant redox properties and play crucial roles in scavenging free radicals and in the breakdown of peroxide. Our results demonstrated that the combination alginate/montmorillonite microcapsules can contribute to the protection of the essential oil against oxidation, leading to the formation of an oxygen barrier and increasing its stability.

### 3.4. REO Release Kinetics

The rosemary essential oil release from the microcapsules depends on the medium of release; the latest penetrates into the microcapsules to dissolve the entrapped essential oil which diffuses through the microcapsules’ membrane. The release studies were performed to show the difference in the release pathway as a function of the microparticle formulation and the REO concentration. The experiment was carried out in 1% Tween 80-water to avoid water saturation due to the low solubility of rosemary essential oil in water. As shown in Figure 7*,* two phases were discerned: an initial phase of fast release correlated to the REO adsorbed on or near the surface of the microcapsules. Contrariwise, the second phase corresponds to the slow release of the REO and stays nearly constant over time. At that stage, the release can be due to the diffusion of the REO dispersed into the microcapsules [42]. The profiles suggest that the release rate of microcapsules with higher REO concentrations is slower than that of microcapsules with lower REO concentrations. The amount of REO released from CA microcapsules (Figure 7A) was approximately equal to the amount released from hybrid CA-MTN microcapsules (Figure 7B), but the time of release differed from 28 to 85 h for CA and hybrid CA-MTN, respectively.

To explain the mechanism of the REO release from the microcapsules, the release data CR (%) were fitted to the Korsmeyer and Peppas semi-empirical model. The obtained kinetic parameters are listed in Table 5. Release-kinetics correlation coefficients (R^2^) of both microparticles were close to 0.9. The diffusional exponent (n) ranged between 0.28 and 0.29 for hybrid CA-MTN and CA microcapsules, respectively. These data indicate that the rosemary essential oil release pathway follows a Fickian diffusion, in which the release is governed by a diffusional behaviour. Therefore, the diffusional process of essential oil in microcapsules can be influenced by the presence of clay particles which prevents the movement of the polymer chains and limits the rate of the internal diffusion of the essential oil in the hybrid microcapsules compared to CA microcapsules.

## 4. Conclusions

Based on the previous results, it can be said that the process of encapsulation is a well-established and efficient method to entrap and maintain molecules. It proposes a plethora of benefits and can be achieved by a myriad of techniques whose main objective is to sustain the release of encapsulated components such as essential oils. REO-loaded microcapsules were prepared by o/w emulsions drop wised into a calcium gelling bath. The TGA thermograms show that hybrid microcapsules CA-MTN have higher thermal stability than CA microcapsules due to the presence of the clay. The rate loss of rosemary essential oil is greater in CA than in hybrid CA-MTN microcapsules which proves the high entrapment capacity of the hybrid system. The microcapsule size and morphology analysis have shown that the obtained particles are micrometric and spherical. Concerning the loading capacity and the encapsulation efficiency, the studies indicate that with a higher value of REO concentration, the loading capacity increased while the encapsulation efficiency decreased for both systems. The REO release is three times slower in hybrid CA-MTN microcapsules than in CA microcapsules which suggests that the hybrid CA-MTN have a higher retention capacity due to the wide specific area of the clay. The kinetic release is of a diffusional type and is correlated with the Korsmeyer–Peppas kinetic model. This work’s main result is to establish a clear influence of the clay filling with sodium montmorillonite from eastern Morocco to obtain the hybrid formulation CA-MTN which shows better thermal stability, higher loading capacity/encapsulation efficiency and slower release of rosemary essential oil using the simple and inexpensive method of ionotropic gelation without including any harmful substance in the whole process. These developed microcapsule systems loaded with rosemary essential oil can be used in active food packaging to protect the food from external factors, and in cosmetics and beauty care products as well as in non-eco-toxic pesticides.

## Figures and Tables

**Figure 1 polymers-15-00823-f001:**
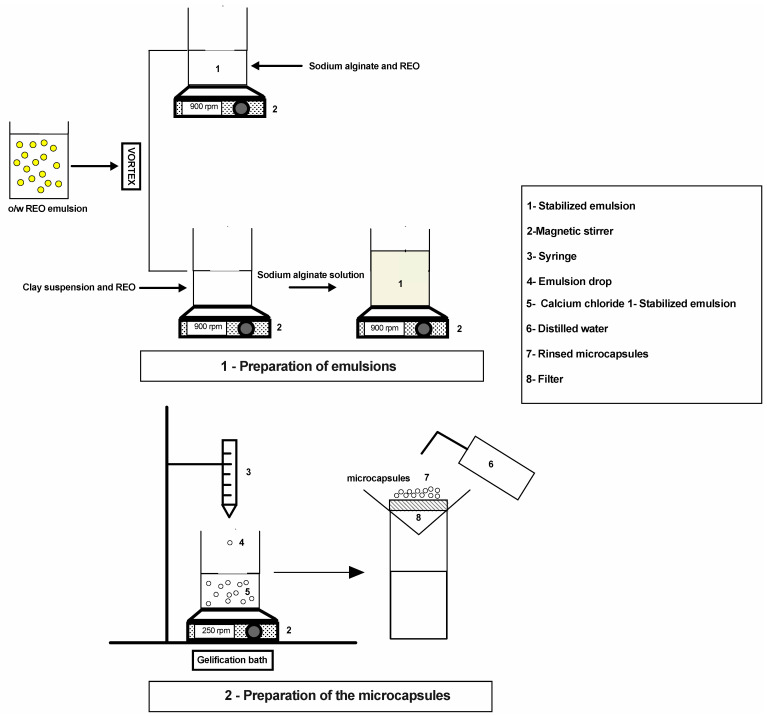
Illustration of the CA and hybrid CA-MTN microcapsules elaboration process.

**Figure 2 polymers-15-00823-f002:**
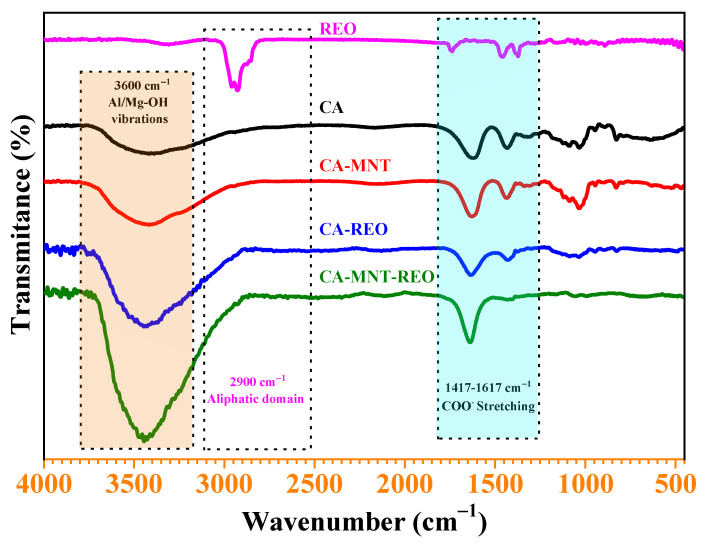
ATR-FTIR spectral of rosemary essential oil (REO), calcium alginate (CA), calcium alginate-montmorillonite (CA-MTN), calcium alginate-rosemary essential oil (CA-REO), and calcium alginate-montmorillonite-rosemary essential oil (CA-MTN-REO).

**Figure 3 polymers-15-00823-f003:**
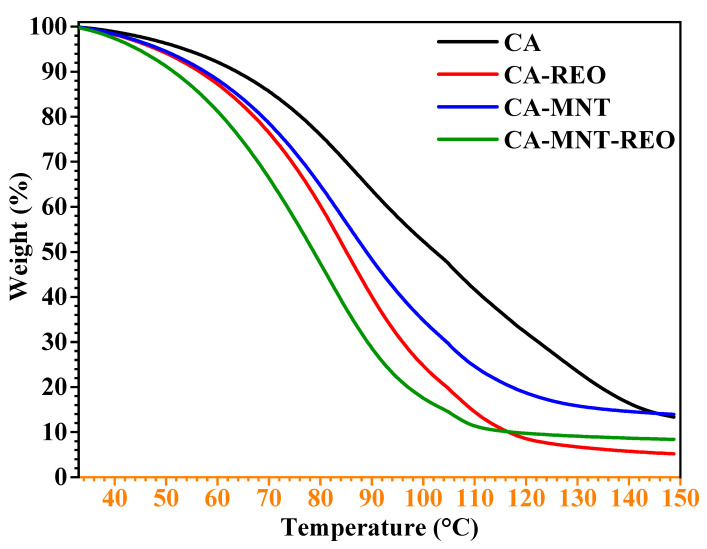
TGA curves of calcium alginate (CA), calcium alginate-montmorillonite (CA-MTN), calcium alginate-rosemary essential oil (CA-REO), and calcium-alginate-montmorillonite-rosemary essential oil (CA-MTN-REO) microcapsules.

**Figure 4 polymers-15-00823-f004:**
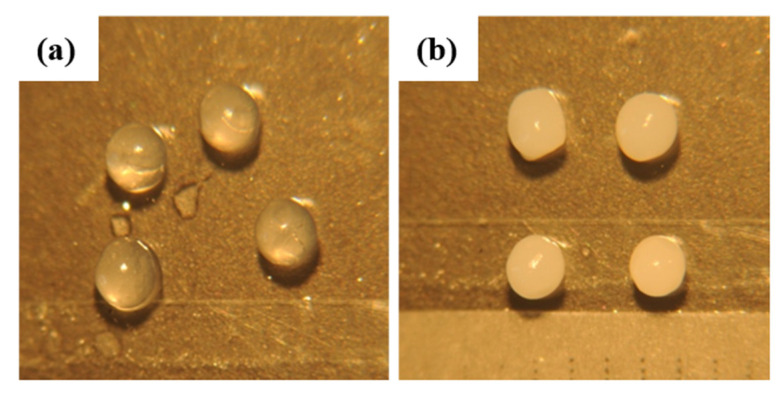
Photograph of CA (**a**) and hybrid CA-MTN (**b**) microcapsules. Magnification ×10.

**Figure 5 polymers-15-00823-f005:**
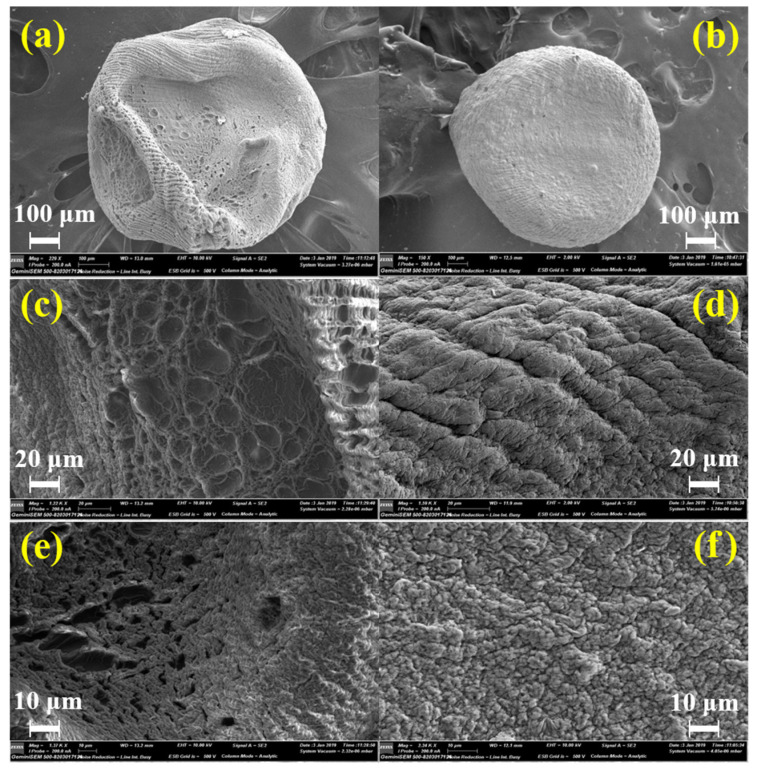
SEM micrographs of CA ((**a**): 229 X, (**c**): 1.22 K X, (**e**): 1.37 K X) and CA-MTN ((**b**): 150 X, (**d**): 1.59 K X, (**f**): 2.24 K X) microcapsules.

**Figure 6 polymers-15-00823-f006:**
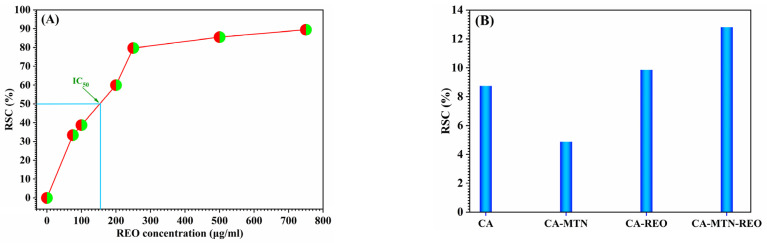
DPPH-scavenging activities of rosemary essential oil (**A**) and of unloaded and REO loaded CA and hybrid CA-MTN microcapsules (**B**).

**Figure 7 polymers-15-00823-f007:**
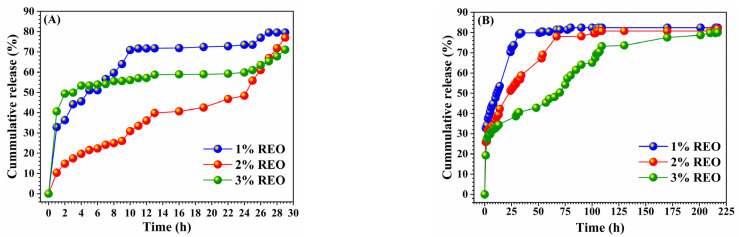
Kinetical release profiles of rosemary essential oil from CA-REO (**A**) and hybrid CA-MTN-REO (**B**) microcapsules in w/o medium.

**Table 1 polymers-15-00823-t001:** The main parameters determined for the raw and homosodic clay. CEC: is the cationic exchange capacity using the copper ethylene diamine complex [Cu (EDA)2]2^+^ and BET: specific surface of clays.

Clay	CEC_EDA-Cu_ (meq/100 g)	BET (m^2^/g)	Pore Volume (cm^3^/g)	Zeta Potential (mV)
Raw	68.74	-	0.009	−27.77
Homosodic	91.66	94.25	0.28	−24.72

**Table 2 polymers-15-00823-t002:** Chemical profile of *Rosmarinus officinalis* L. essential oil (R_T_ is the retention time).

Name of the Compound	R_T_	Name of the Compound	R_T_
α-Pinene	4.983	Camphor	8.308
Camphene	5.225	Borneol	8.625
β-Pinene	5.658	4-Terpineol	8.775
β-Myrcene	5.817	α-Terpieol	8.967
α-Terpinen	6.258	Isobornyl acetate	10.325
m-Cymene	6.383	Copaene	11.600
D-Limonene	6.458	Caryophyllene	12.217
1.8-Cineol	6.525	α-Humulene	12.650
γ-Terpinen	6.917	Isoledene	12.900
cis-β-Terpineol	7.067	γ-Murolene	13.383
Terpinolene	7.383	δ-Cadinene, (+)	13.475
Linalool	7.517	Caryophyllene oxide	14.283

**Table 3 polymers-15-00823-t003:** Size and morphology parameters of CA and hybrid CA-MTN microcapsules.

Microcapsule	CA Microcapsules	CA-MTN Hybrid Microcapsules
Diameter (mean ± SD) (mm)	1.73 ± 0.01	1.64 ± 0.05
Circularity (mean ± SD)	0.92 ± 0.01	0.94 ± 0.01
Aspect Ratio (mean ± SD)	1.05 ± 0.01	1.17 ± 0.06

**Table 4 polymers-15-00823-t004:** Loading capacity (LC) and encapsulation efficiency (EE) for CA and hybrid CA-MTN microcapsules.

Microcapsule	CA Microcapsules			CA-MTN Hybrid Microcapsules		
REO Concentration	1%	2%	3%	1%	2%	3%
LC (%)	26.62 ± 0.19	49.93 ± 0.03	71.36 ± 0.1	28.58 ± 0.21	54.83 ± 0.12	73.6 ± 0.14
EE (%)	92.73 ± 0.14	81.946 ± 0.04	81.9 ± 0.04	83.58 ± 0.23	94.59 ± 0.12	83.59 ± 0.13

**Table 5 polymers-15-00823-t005:** Release kinetic parameters (n and k) for the CA and hybrid CA-MTN microcapsules; n is the release exponent related to the drug release mechanisms, k is the rate constant and R^2^ is the coefficient of correlation.

Kinetic Parameters	CA-REO Microcapsules	CA-MTN-REO Hybrid Microcapsules
n	0.29	0.28
k	0.18	0.12
R^2^	0.93	0.90

## Data Availability

Not applicable.

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
