# Peer review of "Hybrid Microcapsules for Encapsulation and Controlled Release of Rosemary Essential Oil"

_polymers, 2023, doi:10.3390/polym15040823_

Round 1

Reviewer 1 Report

The paper contains information on a new type of "smart carriers" based on calcium alginate/montmorillonite. The paper is interesting for readers and can be accepted for publication as soon as all issues should be clarified.

It will be valuable to illustrate the process of the fabrication of the hybrid microcapsules in the form of the figure.

Please add the figure where the schematic picture of the hybrid microcapsules will be depicted.

I don't find much information in the conclusion about what exactly the differences between CA and CA-MTN microcapsules are.

Figure 4 is poorly described, please add an appropriate explanation why these structures have different shapes.

The quality of figures 5 and 6 should be improved.

Please add a discussion on the application of hybrid microcapsules with rosemary essential oil.

If the developed system will be recommended for advanced biomedical applications, REO release kinetics for buffer solutions should be studied.

I suggest citing papers where similar advanced "smart carriers" were considered:

https://doi.org/10.3390/polym14194245

https://doi.org/10.1016/j.bioactmat.2022.12.002

Author Response

Dear Reviewers,

 We have the pleasure of sending you the modified version of the manuscript entitled “Hybrid microcapsules for encapsulation and controlled release of rosemary essential oil” to be considered for publication as a research article in your journal –Polymers.

Thank you for giving us the opportunity to improve our manuscript by the revised version and thank for your useful comments.

We really appreciate the Reviewers’ comments. We hope this revision will satisfy reviewers' inquiries, and that our work will be considered for publication in your journal.

With kind regards

Dr. Addi and the co-Authors

REVIEWER 1

REVIEWER-1: It will be valuable to illustrate the process of the fabrication of the hybrid microcapsules in the form of the figure. Please add the figure where the schematic picture of the hybrid microcapsules will be depicted.

AUTHORS: Thank you for this suggestion. We have included the figure illustrating the process of the organic and hybrid microcapsules elaboration as figure 1 in the text.

REVIEWER-1: I don't find much information in the conclusion about what exactly the differences between CA and CA-MTN microcapsules are.

AUTHORS: Thank you, we highlighted the difference between both of them in the conclusion

REVIEWER-1: Figure 4 is poorly described, please add an appropriate explanation why these structures have different shapes.

AUTHORS: Thank you for your remark. We added a discussion on the different shapes between the structures of the CA and CA-MTN microcapsules.

REVIEWER-1: The quality of figures 6 and 7 should be improved.

AUTHORS:  Thank you for this comment. The quality of figure 6 and 7 has been improved.

REVIEWER-1: Please add a discussion on the application of hybrid microcapsules with rosemary essential oil.

AUTHORS: Thank you for your remark. The applications concerning the encapsulation of rosemary essential oil specifically into hybrid systems to form capsules like the one we studied are few, not to say almost rare. Mostly we find powdery formulations, nanotubes, films…etc. However, we cited the most recent works and the closest to what we studied.

REVIEWER-1: If the developed system will be recommended for advanced biomedical applications, REO release kinetics for buffer solutions should be studied.

AUTHORS: Thank you for this useful comment. The rosemary essential oil release should be studied according to the aimed application. If the microcapsules are developed for oral taking, it should be studied in a buffer which properties are close to the physico-chemistry of the stomach, if injected, the one of the bloodstream…etc.

In our work, we were more focused on the influence of the formulation on the release behavior of the essential oil because we used two more different types of clay. So we studied the release in a neutral pH, aqueous medium at ambient room temperature.

REVIEWER-1: I suggest citing papers where similar advanced "smart carriers" were considered:
https://doi.org/10.3390/polym14194245,   https://doi.org/10.1016/j.bioactmat.2022.12.002

AUTHORS: The references have been added thanks to your suggestion.

Thank you for all your pertinent comments and suggestions, the manuscript has been modified according to your reviews and it is under track change, so you can find all the modifications that were made. 

Reviewer 2 Report

Hybrid microcapsules for encapsulation and controlled release 2 of rosemary essential oil

The manuscript has been written and presented well, however, the following points must be addressed by authors

Line 42

Despite the great biological potential that essential oils currently possess, they are un- 42 derutilized when their applications become difficult due to certain….

Please complete the sentence.

Line 82

However, one of the challenges in developing a controlled delivery system is the 82 low efficiency of trapping water-soluble substances within alginate [15] .

Please elaborate why hydrophilic loading efficiency is low for hydrophilic alginates?

And how hydrophobic clays will enhance the loading of hydrophilic materials?

Please cite relevant references.

Line 87

The applicability of essential oil encapsulation in a unique or synergistic has proven its benefits 88 due to their antimicrobial, antifungal, anti-inflammatory effects…e

Please cite relevant references.

Line 118

Table 1:The main parameters determined for the raw and homosodic clay

Please add legends and define all symbols / units.

Have you evaluated all parameters in-house or cite if its reported?

Line 124

REO-loaded microcapsules were prepared in a two-step process: In a first time, a sodium 124 alginate and rosemary essential oil emulsion was prepared and then it was drop wised 125 into a gelation bath of calcium chloride.

Please rephrase and correct mistakes.

Line 143

analysis of Rosmarinus officinalis L essential oil was performed using a Hewlett Packard 6890 gas chromatograph equipped with a mass selective detector, i

Please rephrase and correct Please cite reference for this  method

Line 22

The essential oil of Rosmarinus officinalis L. from the eastern region of Morocco has been 228 performed by gas chromatography-mass spectroscopy (GC-MS). A

Please cite the reference of this method.

Line 366

The demand for natural antioxidants in the food industry is growing daily, as well as a 366 growing trend for consumers seeking healthier eating habits. However, monoterpenes are 367 significant antioxidants but volatile flavor ingredients in essential oils

Please rephrase.

Line 387

Figure 5: DPPH-scavenging activities of rosemary essential oil (A) and of REO unloaded and loaded 387 microcapsules(B)

Please improve figure quality.

Line 420

Figure 6: Kinetical release profiles of REO from CA and CA-MTN microcapsules in aqueous medium

Please improve figure quality.

Line 431

Table 5: Release kinetic parameters (n and k) for the CA and CA-MTN microcapsules.

Add legends and define all parameters

Author Response

Dear Reviewers,

 We have the pleasure of sending you the modified version of the manuscript entitled “Hybrid microcapsules for encapsulation and controlled release of rosemary essential oil” to be considered for publication as a research article in your journal –Polymers.

Thank you for giving us the opportunity to improve our manuscript with the revised version and thank you for your useful comments.

We really appreciate the Reviewers’ comments. We hope this revision will satisfy reviewers' inquiries, and that our work will be considered for publication in your journal.

With kind regards

Dr. Addi and the co-Authors

-------

REVIEWER 2

REVIEWER-2: Line 42, Despite the great biological potential that essential oils currently possess…. Please complete the sentence.

AUTHORS : Thank you for your remark, the sentence has been completed.

REVIEWER-2: Line 82, However, one of the challenges developing…. Please elaborate why hydrophilic loading efficiency is low for hydrophilic alginates? And how hydrophobic clays will enhance the loading of hydrophilic materials.

Please cite relevant references.

AUTHORS: Thank you for this relevant comment. Actually, the paragraph wasn’t well written and was mistaken. The whole paragraph has been deleted. Thank you.

REVIEWER-2: Line 87, The applicability of essential oil encapsulation in a unique or synergetic… Please cite relevant references.

AUTHORS: Thank you for the suggestion. The references for the paragraphs have been added.

REVIEWER-2: Line 118, Table 1: The main parameters determined for the raw and homosodic clay. Please add legends and define all symbols/units.

Have you evaluated all parameters in house or cite if reported.

AUTHORS: Thank you. The legends have been added to the table1.

Yes, the clay has been already evaluated and the paper reporting the results is already cited before the table 1 in the materials section.

REVIEWER-2: Line 124, REO loaded microcapsules were prepared in a two-step process… Please rephrase and correct mistakes.

AUTHORS: Thank you for the suggestion, the lines have been rephrased and mistakes have been corrected.

REVIEWER-2: Line 143, Analysis of Rosmarinus officinalis L. essential oil… Please rephrase and correct. Please cite reference for this method.

AUTHORS: Thank you very much, the paragraphs have been rephrased and corrected. A reference for the method has been cited.

REVIEWER-2: Line 22, The essential oil of Rosmarinus officinalis L. from the eastern region of Morocco…Please cite reference for this method.

AUTHORS: Thank you for the suggestion. The reference for the method has been added.

REVIEWER-2: Line 366, The demand for natural antioxidants… Please rephrase.

AUTHORS: Thank you for your comment. The paragraph doesn’t add any relevant information. It has been removed.

REVIEWER-2: Line 387, Fig 6. Please improve the quality.

AUTHORS: Thank you. The figure quality has been improved.

REVIEWER-2: Line 420, Fig 7 Please improve the quality.

AUTHORS: Thank you. The figure quality has been improved.

REVIEWER-2: Line 431, Table 5, Release kinetic parameters n and k for the CA and CA-MTN microcapsules. Add legends and define all parameters.

AUTHORS : Thank you for your suggestion. The legends and definitions have been added.

Thank you for all your pertinent comments and suggestions, the manuscript has been modified according to your reviews and it is under track change, so you can find all the modifications that were made.

Reviewer 3 Report

This work is devoted to hybrid microcapsules for encapsulation and controlled release of rosemary essential oil. The microcapsules were developed based on calcium alginate and calcium alginate/montmorillonite following the ionotropic crosslinking gelation. The unloaded/loaded CA and CA-MTN microcapsules were characterized by FT-ATR spectroscopy, thermal analysis, scanning electron microscopy and DPPH assay. It was shown that the clay-alginate combination provides microcapsules with enhanced properties compared to the polymer-based microcapsules. Taking into account the mentioned below notes, I think that the article looks like a short communication and may be published after minor revision.

Notes:

1. The meaning of abbreviation “DPPH” should be written at the first mentioning in the text (excepting Abstract).

2. The abbreviation of “Attenuated Total Reflectance-Fourier Transform Infrared (ATR-FTIR)” should be written once at the first mentioning.

3. In the description to Figure 4 the c), d), e), f) should be mentioned.

4. Are the developed by the authors calcium alginate/montmorillonite (CA-MTN) hybrid microcapsules suitable for all essential oils? Are there any restrictions?

5. In the conclusions, the authors should still summarize the results and write about the findings of this work, and not indicate the main aim of the work. Moreover, promising application of new obtained results should be reflected here.

Author Response

Dear Reviewers,

 We have the pleasure of sending you the modified version of the manuscript entitled “Hybrid microcapsules for encapsulation and controlled release of rosemary essential oil” to be considered for publication as a research article in your journal –Polymers.

Thank you for giving us the opportunity to improve our manuscript by the revised version and thank you for your useful comments.

We really appreciate the Reviewers’ comments. We hope this revision will satisfy reviewers' inquiries, and that our work will be considered for publication in your journal.

With kind regards

Dr. Addi and the co-Authors

REVIEWER 3

REVIEWER-3: The meaning of abbreviation “DPPH” should be written at the first mentioning in the text (excepting Abstract). 

AUTHORS: Thank you, the meaning of the abbreviation “DPPH” 1.1-diphenyl-2-picrylhydrazyl has been written

REVIEWER-3: The abbreviation of “Attenuated Total Reflectance-Fourier Transform Infrared (ATR-FTIR)” should be written once at the first mentioning.

AUTHORS: Thank you, the abbreviation of “Attenuated Total Reflectance-Fourier Transform Infrared (ATR-FTIR)” has been written

REVIEWER-3: In the description to Figure 4 the c), d), e), f) should be mentioned. 

AUTHORS: Thanks to your suggestion, the description to Figure 4 the c), d), e), f) has been mentioned. 

REVIEWER-3: Are the developed by the authors calcium alginate/montmorillonite (CA-MTN) hybrid microcapsules suitable for all essential oils? Are there any restrictions?

AUTHORS: Thank you for your relevant comment, no there are no restrictions, essential oils possess almost the same properties and the developed system Alginate/montmorillonite is suitable to encapsulate other essential oils and their pure compounds.

REVIEWER-3: In the conclusions, the authors should still summarize the results and write about the findings of this work, and not indicate the main aim of the work. Moreover, promising application of new obtained results should be reflected here.

AUTHORS: Thank you for your suggestion. As suggested by the Reviewer, promising applications and highlighting the results were reflected in the conclusion.

Thank you for all your pertinent comments and suggestions, the manuscript has been modified according to your reviews and it is under track change, so you can find all the modifications that were made.

Round 2

Reviewer 1 Report

The authors have answered all my comments and the paper can be accepted in its present form.